biochemistry/cellular biology/molecular biology

photodynamic therapy, photosensitizer, lung cancer stem cells, cell death, cytotoxicity

**Authors for correspondence:**
Anine Crous
e-mail: acrous@uj.ac.za
Heidi Abrahamse
e-mail: habrahamse@uj.ac.za

# Aluminium (III) phthalocyanine chloride tetrasulphonate is an effective photosensitizer for the eradication of lung cancer stem cells

## Anine Crous and Heidi Abrahamse

Laser Research Centre, Faculty of Health Sciences, University of Johannesburg, PO Box 17011, Johannesburg 2028, South Africa

AC, 0000-0002-7581-7731; HA, 0000-0001-5002-827X

Cancer stem cells (CSCs) are considered to contribute to the recurrence of lung cancer due to their stem-like nature and the involvement of genetic markers associated with drug efflux, regeneration and metastases. Photodynamic therapy (PDT) is a cost-effective and non-invasive therapeutic application that can act as an alternative therapy for lung cancer when considering CSC involvement. Stem-like cells derived from the A549 lung cancer cell line, positive for CD133, CD56 and CD44 antigen markers, were characterized, intracellular localization of aluminium (III) phthalocyanine chloride tetrasulphonate (AlPcS$_4$Cl) determined and its anti-cancer PDT effects were evaluated. Results confirmed that isolated cells were stem cell-like and subcellular localization of AlPcS$_4$Cl in integral organelles involved in cell homeostasis supported the destruction of CSC. AlPcS$_4$Cl's effectivity was demonstrated with CSC eradication showing a significant increase in cytotoxicity and cell death via apoptosis, caused by a decrease in mitochondrial membrane potential. PDT could serve as a palliative treatment for lung cancer and improve prognosis by elimination of lung CSCs.

## 1. Introduction

Research findings suggest that a small heterogeneous subpopulation of cells is the origin of cancer development [1]. This small population of cells called cancer stem cells (CSCs) display stem-like characteristics. CSCs can self-renew, have

multi-differentiation potential, can metastasize [2] and are known to evade drug-induced cell death, mainly due to their quiescent stem-like nature [3], where this state of dormancy is characterized as a recurrent clinical occurrence where diffuse tumour cells are retained in a non-proliferating quiescent state for long periods of time. This phenomenon can occur at an early stage of the disease or after a therapeutic intervention. The activation of these dormant cells contributes to tumour growth and relapse [4,5]. Self-conservation of CSCs is allowed by asymmetrical cell division cycles, where the CSC population is retained resulting in a heterogeneous tumour population of CSCs and non-stem-like cancer cells [6]. These non-stem-like cancer cells undergo rapid symmetrical cell proliferation, making them susceptible to conventional cancer treatment and preserving the CSC population, justifying treatment failure and cancer relapse [7]. Such significant clinical findings further prompted a strong interest in alternative methods for further study of CSCs and their involvement in the management of drug-resistant lung cancer. CSCs demonstrate high drug resistance and resilience due to extended telomere duration, initiation toward apoptotic pathways, increased membrane transporter activity and their ability to travel and metastasize [8].

Lung cancer is a noteworthy supporter of disease fatalities [9]. Patients undergoing lung cancer treatment can still only expect a 5-year survival rate due to treatment resistance before and during chemo and radiation therapy. Treatment resistance contributes to disease progression, recurrence and mortality [10]. This poses a significant clinical challenge for lung cancer treatment. Novel agents have been used alone and in combination with conventional therapy to overcome this problem. However, the basic systems presenting this resistant phenotype in malignant lung growth stay obscure [11]. The complete treatment of cancer depends on revealing its origin. Lung CSCs identified and showed resistance to various lung cancer treatment options. These include conventional therapy, biological molecules and targeted therapy. Eliminating lung CSCs during therapeutic intervention is of utmost importance as this can prevent CSC expansion, cancer recurrence, relapse and metastasis. While little is at present known about lung CSC biology, various CSC markers have been distinguished and considered. These markers incorporate but are not limited to ALDH1, CD133, side population (Hoechst-negative), CD44, CD87 and CD117. These markers have been connected to chemoresistance in various first-line disease treatments [11]. It is therefore widely accepted that CSCs are closely related to pathological features resulting in poor clinical prognosis [1].

Photodynamic therapy (PDT) or alternative/photochemotherapy is a non-intrusive, innovative form of cancer treatment typically carried out as an outpatient procedure that can be used in isolation or in conjunction with traditional cancer treatments [12]. The mechanism of action for PDT comprises the photosensitive compound being activated through photon absorption from a specific light wave. The activated photosensitiser (PS) generates reactive oxygen species (ROS) leading to cell membrane damage and cell death [13]. Despite PDT achieving high success rates in treating various cancers, researchers are investigating the improvement of PDT by enhancing its selectivity and effectiveness by developing new PS compounds and improving PS delivery methods [14]. Aluminium (III) phthalocyanine chloride tetrasulphonate (AlPcS$_4$Cl) is an improved second-generation PS with ideal PDT characteristics. Phthalocyanines (Pcs) are synthetic dyes that have a high molar absorption coefficient in the red part of the spectrum. Pcs belong to the group of second-generation PSs which exhibit a high extinction coefficient around 670 and 750 nm and even up to 1000 nm, and thus, allowing increased tissue penetration of the activating light [15]. PSs currently used in clinic exhibit high hydrophobicity which limits their clinical efficacy [16]. They have lower triplet state lifetimes and yields and less singlet oxygen production compared with sulphonated AlPc [17]. Porphyrin and a hydrophobic hydroxyphenyl chlorin, most frequently used in clinical settings, have comparatively lower cellular uptake [18] along with suboptimal photochemical characteristics seen when using a hydrophilic form of porphyrin [19]. However, AlPcS$_4$Cl has been modified through axial and peripheral substituents [17] where it has tetrapyrrolic, aromatic macrocycles with each cycle connected to the next by nitrogen particles, each pyrrolic ring is reached out by a benzene ring bringing about the red shift of their last retention band. An electron-donating group, -SR at the non-peripheral and periphery of the PS results in a red shift to the NIR area [20]. In addition, the nearness of a diamagnetic focal metal, Al3+, in the Pc core appears to improve the triplet state lifetime ($\tau$t), just as its yield ($\Phi$t) and singlet oxygen yields ($\Phi\Delta$) contrasted with paramagnetic metals [17]. The PS's sulfonation essentially expands Pc dissolvability in polar solvents including water, evading the requirement for elective conveyance vehicles [15]. AlPcS$_4$ has much better photochemical characteristics for use in PDT ($\varepsilon675 = 1.7 \times 10^5$ versus $\varepsilon595 = 7 \times 10^3\,\mathrm{M}^{-1}\,\mathrm{cm}^{-1}$) [19]. Along with the aforementioned characteristics, AlPcS$_4$Cl shows little to no dark toxicity and is amphiphilic in nature due to it being both water soluble and can bind to cytochrome c located in the mitochondrial membrane [15].

The CSC hypothesis has attracted the development of therapies directed at the destruction of CSCs, as well as the reduction of tumour mass by the treatment of non-stem-like cancer cells. PDT can offer this solution by being used as an alternative to traditional cancer therapy. The innovative use of CSC eradication using AlPcS$_4$Cl for PDT will provide insight into the therapeutic ability of PDT in the treatment of lung cancers. The following table (table 1) illustrates previous laboratory trials in which AlPc-compounds have been successfully used to kill cancer cells from different cell lines and some PSs that have been tested for their effects on different CSCs.

The goal of this research was to assess the efficacy of AlPcS$_4$Cl as a PS in the eradication of lung CSCs. The PDT capability was established by CSC morphology, cytotoxicity, proliferation, viability, mitochondrial membrane potential and cell death mechanism.

# 2. Material and methods

## 2.1. Cell culture

Commercially accessible epithelial lung carcinoma cells from the ATCC were used in this investigation, A549 (ATCC®, CCL185TM). The media used for maintaining the lung cancer cells and isolated CSCs comprised the accompanying: Rosewell Park Memorial Institute 1640 medium (RPMI-1640) (SIGMA, R8758) enhanced with 10% fetal bovine serum (Biochrom, S0615) and 0.5% penicillin/streptomycin (SIGMA, P4333) and 0.5% amphotericin B (SIGMA, A2942). All cultured cells were maintained and incubated at 37°C in 5% CO$_2$ and 85% humidity.

## 2.2. Isolation of cancer stem cells

A magnetic bead isolation pack and division unit (Miltenyi Biotec, QuadroMACSTM partition unit 130-091-051) was used to separate lung CSCs from lung cancer cells. The cells were magnetically labelled with microbead-conjugated antibodies coordinated at the antigenic surface marker CD133, CD56 and CD44. The cell population was enhanced for CD133+, CD56+ and CD44+ lung CSCs by using the CD133/CD56/CD44 Microbead Kit (Miltenyi Biotec, MicroBead Kit, 130-095-194, 130-100-857, 130-050-401) intended for the positive selection of cells expressing the human CD133/CD56 and CD44 antigen. Lung cancer cells that did not express the antigens were eluted and discarded, where positively selected cells were kept and cultured to use for biochemical analysis. The method for positive isolation was carried out using the manufacturers' protocol.

## 2.3. Characterization of isolated lung cancer stem cells

### 2.3.1. Flow cytometry

To confirm whether cells isolated using the magnetic bead kit were of CSC origin, cells were fluorescently labelled using the secondary antibody conjugation technique. Primary mouse anti-human antibody CD133, CD56 and CD44 was used ((CD133 Antibody (3F10) (NovusBio, NBP2-3774)); (CD56 Monoclonal Anti-N Cam (Sigma-Aldrich, C9672)); CD44 Antibody (8E2F3) (NovusBio, NBP1-47386)) and counterstained with secondary fluorescence (FL) antibodies FITC Goat anti-Mouse (NovusBio, NB720-F-1 mg), Cy5 Goat anti-Mouse (NovusBio, NB7602) and PE Goat anti-Mouse (NovusBio, NB7594) [34]. Antigenic detection was confirmed using the BD Accuri™ C6 flow cytometer (BD Biosciences, BD ACCURI C6) which detects the FL on the conjugated antibody, indicating if the cells are CD133, CD56 and CD44 positive/negative.

### 2.3.2. Immuno fluorescence

The isolated CSCs were characterized using indirect FL microscopy. Whereby cultured cells are labelled with a primary antibody directed at a target antigen and made visible using fluorochromes attached to secondary antibodies directed at the primary antibodies. Cells cultured on heat-sterilized coverslips were incubated with primary mouse anti-human antibody (CD133 Antibody; CD56 Monoclonal Anti-N Cam; CD44 Antibody), and fluorescently labelled with FITC-conjugated secondary anti-mouse [34]. Cells were counterstained with 4′-6-diamidino-2-phenylindole (DAPI) (InvitrogenTM, D1306) and viewed using a FL microscope (OLYMPUS CKX41) using the OLYMPUS cellSens Imaging Software. The images were

**Table 1.** AlPc–PDT tested on cancer cell lines and other PSs evaluated on CSCs for their photodynamic effects.

**AlPc–PDT**

| PS | cancer cell line | effect | ref |
|---|---|---|---|
| AlPc-NE | mammary adenocarcinoma (4T1, ATCC® CRL-2539™) | decreased cell viability, increased cytotoxicity; eradication of primary 4T1 tumours and pulmonary metastasis | [21] |
| AlPcS₄Cl | pancreatic adenocarcinoma (BXPC3, Merck 93120816) | excellent photostability, lysosomal permeabilization | [22] |
| AlPcS₄Cl | lung cancer (A549, ATCC® CCL-185) | dose depended increase in cell death; increase in cytotoxicity and decrease in proliferation and viability | [23] |
| AlPcS$_{mix}$ | cervical cancer (HeLa; HPV18, ATCC® CCL2™) | dose-dependent response; increase in cell toxicity; decrease in viability and proliferation | [24] |
| AlPcS₄Cl - 1,8,15,22-tetrakis (-phenylthio)-29H,31H | gastric cancer (EPG85-257P, CVCL_X333; EPG85-257RDB, CVCL_X175) | decreased viability | [25] |
| AlPc-NE | breast adenocarcinoma (MCF7, ATCC® HTB-22™) | dose-dependent ROS production; increased cytotoxicity and decreased cell viability | [26] |
| AlPc-mMAb 425 | vulvar SCC (A431) | increased cytotoxicity | [27] |

**CSC–PDT**

| PS | CSCs | effect | ref |
|---|---|---|---|
| AlPcS₄Cl – AuNP and AlPcS₄Cl – AuNP – CD133 | lung CSCs (A549, ATCC® CCL-185) | significant CSC toxicity and cell death; CSC eradication | [28] |
| Rose Bengal | colon CSCs (Lgr5-EGFP-IRES-creERT2) | eradication of Lgr5+ CSCs *in situ* | [29] |
| chlorin e6 – SWCNT – HA | colon CSCs (Caco-2, ATCC® HTB-37TM) | increased cytotoxicity and apoptotic cell death | [30] |
| verteporfin | gastric CSCs (MKN45 and MKN74 cell lines); (GC04, GC06, GC07, GC10, GC44 and GC35 patient-derived GC) | inhibited Y/T-TEAD transcriptional activity, cell proliferation and CD44 expression; Inhibited tumour growth *in vivo* and tumour sphere formation *in vitro* | [31] |
| 5-ALA | pancreatic CSCs (PANC-1, ATCC® CRL-1469™) | no effect on sphere forming ability | [32] |
| AlPcS$_{mix}$ | cervical CSCs (HeLa; HPV18, ATCC® CCL2™) | dose-dependent increase in cell toxicity, decrease in viability and proliferation | [24] |
| lovastatin (LV) photosan-II | nasopharyngeal carcinoma CSCs (5-8F and 6-10B, Cancer Research Institute of Central South University) | LV Inhibit proliferation and stemness, increased cell death and sensitivity to cyclophosphamide; LV+PDT dose-dependent decrease in viability, caused cytotoxicity | [33] |

**Figure 1.** Structural components of AlPcS$_4$Cl, comprising a porphyrin ring, pyrolic/benzene rings and sulfate functional groups attached to a metalized phthalocyanine.

compiled using a Java image processing program, ImageJ, developed at the National Institutes of Health and the Laboratory for Optical and Computational Instrumentation, Licence: Public Domain, BSD-2.

## 2.4. Localization of aluminium (III) phthalocyanine chloride tetrasulphonate in intracellular organelles of lung cancer stem cells

Localization of AlPcS$_4$Cl into lung CSC cellular organelles was confirmed by FL and differential interference contrast (DIC) microscopy, to establish the suitability of AlPcS$_4$Cl as a PS for PDT. Intracellular organelles such as mitochondria and lysosomes were fluorescently labelled. Cells received AlPcS$_4$Cl at a concentration of 20 µM which was used as the treatment concentration. The cells were incubated for at least 4 h allowing the PS to penetrate the cells. Cells were then stained with MitoTracker™ Green FM 490Ex/516EM (Invitrogen™, M7514) for mitochondria [35] or LysoTracker™ Green DND-26 504Ex/511EM (Invitrogen™, L7526) for lysosomes [36]. Cells were counterstained using DAPI. Cells were examined using the live-cell station from Zeiss (Zeiss, Axio Observer Z1) to determine the localization of the AlPcS$_4$Cl.

## 2.5. Photodynamic therapy

### 2.5.1. Photosensitizer

The PS AlPcS$_4$Cl (Frontier Scientific, AlPcS-834) was used in this study. Its formula is C$_{32}$H$_{16}$AlClN$_8$O$_{12}$S$_4$, formula weight 895.21, thin-layer chromatography (TLC) greater than 95% with an absorbance wavelength of 674 nm. Figure 1 shows the structural components of AlPcS$_4$Cl. A predetermined concentration of the PS was used as established in a previous study conducted using AlPcS$_4$Cl on A549 lung cancer cells where a concentration of 20 µM delivered an inhibitory concentration of 50% [IC50] along with a predetermined laser energy output of 10 J cm$^{-2}$ [23], in order to obtain comparative results and perform downstream applications on the CSCs.

### 2.5.2. Laser irradiation

A 673.2 nm diode laser (Arroyo, high power laser, 1000 mA Laser source 4210 (S/N 070900108)), provided by the National Laser Centre of South Africa, was used to irradiate the cells. Prior to exposing the cells to irradiation, the power output of the laser was measured using a FieldMate laser

**Table 2.** PDT drug and laser parameters using the 673.2 nm diode laser.

| parameters | |
| --- | --- |
| laser type | semiconductor (diode) |
| wavelength (nm) | 673.2 |
| wave emission | continuous |
| fluence (J cm$^{-2}$) | 10 |
| photosensitiser (PS) | AlPcS$_4$Cl |
| PS concentration (µM) | 20 |

power meter (FieldMate, Coherent, Power Sens detector (0496005)). The value from the meter reader was used to calculate the exposure time. Laser parameters are seen in table 2.

The laser is set up with a fibre optic to deliver red light onto the monolayer of cells. The fibre optic is placed 8 cm above the cell culture dish, giving an irradiation spot size of 9.1 cm$^2$ covering the entire area of the culture dish. Cells were irradiated without the culture dish lid on at room temperature. To eliminate light interference, all irradiation protocols were performed in the dark. The following calculation was used:

$$mW\,cm^{-2} = \frac{mW}{\pi \times r^2},$$
$$W\,cm^{-2} = \frac{mW\,cm^{-2}}{1000},$$
$$Time\ (s) = \frac{J\,cm^{-2}}{W\,cm^{-2}}.$$

Cells were cultured at a total of $5 \times 10^5$ cells per Petri dish in complete media. Cultures were divided into four study groups. Group 1 was the control and received no irradiation or PS, group 2 contained PS but no irradiation and group 3 was irradiated but no PS was added. Group 4 was treated with AlPcS$_4$Cl and was irradiated. All samples were incubated for 24 h after PDT treatment followed by biochemical analysis.

### 2.5.3. Morphology

Any morphological changes of the isolated lung CSCs post-PDT treatment were observed and studied using an Olympus CKX41 inverted light microscope (Wirsam, Olympus CKX41) 24 h post-irradiation, whereby changes were captured using the SC30 Olympus camera.

### 2.5.4. Cytotoxicity

To determine the toxicity induced by PDT on the CSCs, the amount of lactate dehydrogenase (LDH) released from the cells due to membrane damage and cell death was measured. The CytoTox96® non-radioactive cytotoxicity assay (Promega, G400) was used according to manufacturer specifications [37]. We evaluated cytotoxicity by measuring formazan with a multilabel counter (Perkin Elmer, VICTOR3™, 1420) at 490 nm.

### 2.5.5. Proliferation

The effect of PDT on CSC metabolism was determined by measuring the amount of ATP present in the cells after treatment. Intracellular ATP was measured using the CellTiter-Glo® luminescent cell proliferation assay (Promega, G7570) and used according to manufacturer specifications [38]. Cellular ATP producing a luminescent signal was read and quantified using a multilabel counter (Perkin Elmer, VICTOR3™, 1420) expressed in relative light units (RLU).

### 2.5.6. Viability

The adjustment in cell numbers because of development restraint instigated by PDT on the CSCs was resolved using the trypan blue viability assay. This colour prohibition test permits live (feasible) cells

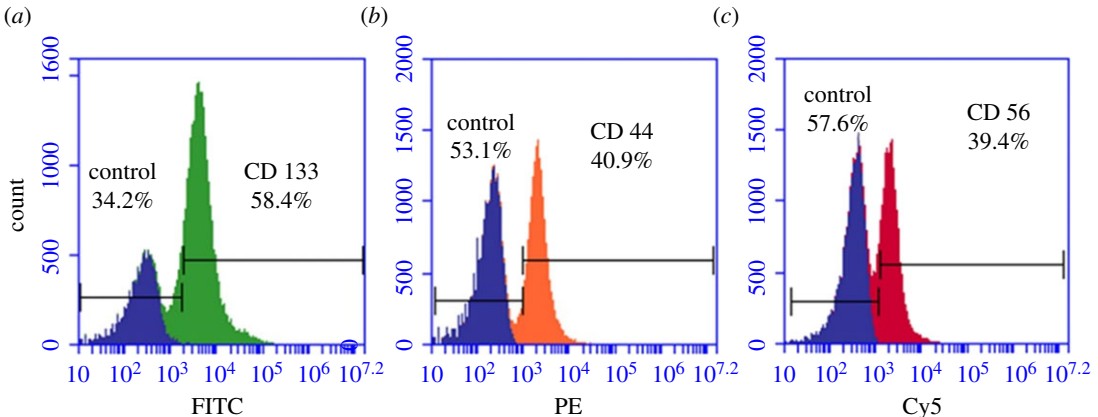

**Figure 2.** Flow cytometry lung CSC characterization. FL protein detection using flow cytometry. (*a*) Lung CSCs positive for CD133 (FITC), (*b*) lung CSCs positive for CD44 (PE) and (*c*) lung CSCs positive for CD56 (Cy5). All positive samples are overlaid with the control to distinguish between the colour shifts.

with flawless membranes to bar the colour and stay unstained, though dead (non-suitable) cells will hold the colour and stain blue. An equivalent volume of cells and 0.4% Trypan blue (Invitrogen™, T10282) was homogeneously blended and exchanged to a chamber slide, which was then embedded into a computerized cell counter (Countess® Automated Cell Counter), which visually delineates the cells and after that electronically calculates the percentage of viable cells.

### 2.5.7. Mitochondrial membrane potential

MitoRed (Mito Red) (Sigma-Aldrich, 53271), a vital dye and mitochondrial stain, was used to indicate the status of mitochondria in the lung CSCs after irradiation. This cell membrane permeable rhodamine-based dye is found in mitochondria and emits red FL. The interaction of Mito Red with mitochondria reflects on the potential of its membrane.

### 2.5.8. Cell death mechanism

Annexin V PI assay kit (BD Pharmingen™, 559763) was used to identify the cell death mechanism induced by AlPcS$_4$Cl-PDT. The assay gives an indication of cell death either as necrotic or apoptotic. The test makes use of fluorescently labelled dyes which are then read using a flow cytometer quantifying the amount of fluorescently labelled cells. Flow cytometric analysis was performed on the BD flow cytometer (BD Accuri™ C6 Cytometer), and Annexin V-FITC and propidium iodide (PI) were detected as a green and red FL, respectively [39].

## 2.6. Statistics

All quantitative experiments were repeated three times. Data processing was done using Sigma plot version 12/13. Error bars are representative of the mean (s.e.m.) (*n* = 3). Data accumulated was statistically evaluated by Student's paired t-test, and the significance was defined as $p < 0.05$ (*), $p < 0.01$ (**) or $p < 0.001$ (***). All colorimetric, luminescent and absorbance assays was performed in duplicate and using a background control that was subtracted from the raw data obtained.

# 3. Results

## 3.1. Flow cytometry

To determine whether the cells isolated were of stem cell origin, antigenic detection was assessed using the BD Accuri™ C6 flow cytometer. Indirect antibody labelling allowed to fluorescently detect the antigen–antibody conjugation. Fluorescent detection using flow cytometry showed that the isolated cells expressed the markers CD133, CD44 and CD56 (figure 2).

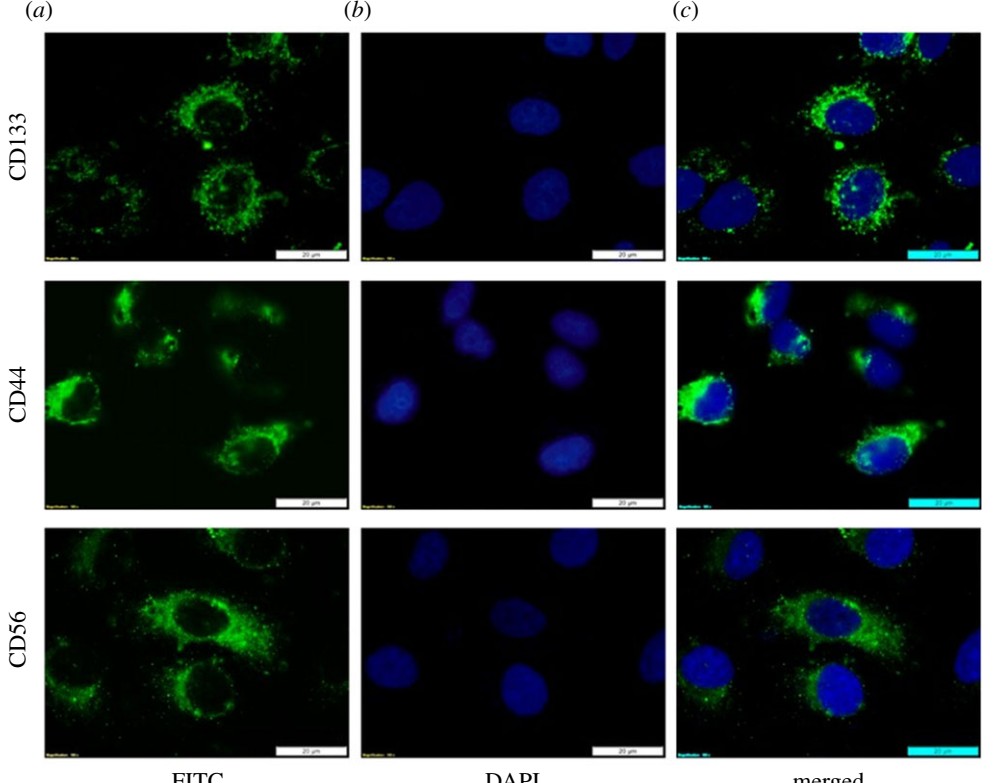

**Figure 3.** Immuno fluorescent lung CSC characterization. Fluorescent antigenic detection of the surface markers CD133, CD44 and CD56. (*a*) Immunofluorescent staining of the isolated side population of lung CSCs using FITC. (*b*) Cells counterstained with DAPI indicated by blue FL seen in the nuclei. (*c*) Superimposed fluorescent images of the labelled cells.

## 3.2. Immuno fluorescence

The expression of the antigenic CSC markers after isolation of the side population was determined by FL microscopy using indirect immune staining. Isolated lung CSCs expressed the surface markers CD133, CD44 and CD56. DAPI was used to counterstain the nuclei. Positive expression of the antigenic surface markers confirmed positive isolation of lung CSCs (figure 3).

CSCs have been isolated from solid tumours and cell lines and their intense ability to initiate tumour formation as well as differentiate upon *in vivo* application has been demonstrated [40]. Studies on how to identify and isolate CSCs have elaborated on the various markers, gene expression molecules and cell signalling pathways involved in CSC research. The most common method applied in identifying and isolating subpopulations of cells is through protein marker detection or cluster of differentiation molecules. Overexpression of common stem cell genes or markers representative of the tissue of origin has been seen in various stem cells including CSCs. Allowing for the identification of stemness inside a tumour population [41]. One way of differentiating CSCs from non-tumorous stem cells and cancerous cells is that CSCs undergo altered glycosylation during malignant transformation. Giving CSCs cancer-specific glycans and their tumorous characteristics of tumour aggressiveness, progression and metastasis along with increased levels of stem marker expression [42]. Thus, the proven successful isolation of the lung CSCs enabled us to further investigate whether treating these cells using PDT could be effective bearing in mind their characteristics of drug evasion attributed to their specific CSC marker expression.

## 3.3. Localization of aluminium (III) phthalocyanine chloride tetrasulphonate in lung cancer stem cells

The intracellular confinement of a PS is critical to characterize the system and effectiveness of photoactuated cell death. Mitochondria are associated with cell homeostasis by directing ATP levels just as framing synthetic substances for the breakdown of poisons, they contain different atoms including DNA, ribosomes and compounds for protein and phospholipid combination [43]. Mitochondria have recently become an attractive target for anti-cancer therapy. Mitochondria have a vital role to play in controlling

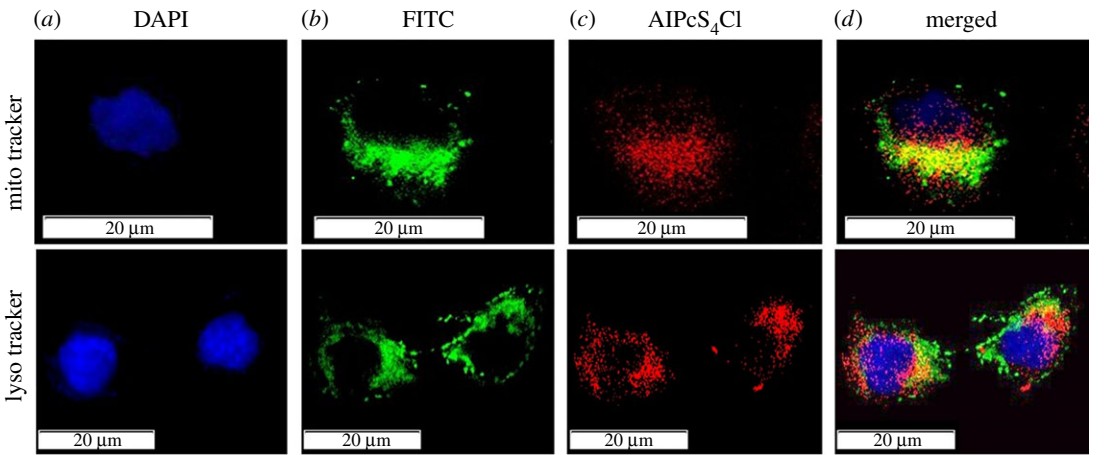

**Figure 4.** Intracellular localization of AlPcS$_4$Cl in isolated lung CSCs. (*a*) Nuclei are counterstained with DAPI, (*b*) mitochondria/lysosomes fluoresces green (FITC), (*c*) AlPcS$_4$Cl auto fluoresces red (DsRED) and (*d*) intermediate yellow is seen in the superimposed images where the green and red channels are merged and FL from the mitochondrion/lysosomes and PS are overlapping.

CSC identity differentiation and survival [44]. Similar mitochondrial functions were reported to help the development and maintenance of the CSC phenotype [45,46]. Focusing on the mitochondria as a PDT target will prompt CSC destruction. Lysosomes contain proteins for hydrolysis or processing of biomolecules just as outside particles to help in the expulsion of atoms from the cell. In the case of CSCs, lysosomal function includes maintaining an acidic environment that aids in CSC invasiveness and resistance to anti-cancer therapies [47]. Impairing lysosomal function can prompt downregulation of cell homeostasis and cell death. To establish the suitability of AlPcS$_4$Cl as a PS we determined where in the CSC AlPcS$_4$Cl localizes and if it localizes in intracellular organelles such as mitochondria and lysosomes. Localization was confirmed using FL microscopy, where mitochondria and lysosomes were fluorescently labelled using FITC and the PSs' auto FL exploited and detected using FL microscopy (figure 4).

Results show AlPcS$_4$Cl is embedded in the cytosol and perinuclear region that upon photoactivation can cause peripheral photodamage to the cell membrane and plasma membranes of various intracellular organelles including mitochondria and lysosomes. Endocytosis is noted where the PS is dispersed throughout the cytosol where, after photoactivation, the PS is released from the endocytic vesicles, ROS, mostly singlet oxygen, is produced and the constituents of the membranes are damaged [48]. This signifies those organelles involved in cell function and viability would be targeted for photoactivation of AlPcS$_4$Cl, after which the development and accumulation of ROS or singlet oxygen may destroy these organelles and can contribute to the death of CSC.

## 3.4. Photodynamic therapy

To determine whether the PS AlPcS$_4$Cl exerts any responses in isolated lung CSCs, an array of biochemical assays was tested along with morphological features. This gives an indication of the extent of PDT damage by assessing post-irradiation cellular phenotypic components. A previously established [IC50] was used where a dose–response of AlPcS$_4$Cl-PDT was examined on the parent lung cancer cell line (table 3).

### 3.4.1. Morphology

Following PDT on the isolated lung CSCs, the cells were characterized morphologically. All the treatment groups were compared with the respective control sample which did not receive any treatment. The results (figure 5) indicated that the isolated lung CSCs receiving irradiation alone had an increase in cell proliferation indicated by an increase in the monolayer density. This is in accordance with previous studies conducted using photo-biomodulation of 10 J cm$^{-2}$ and wavelengths ranging from 630 to 830 nm on lung CSCs and various cancer cell lines, having an increase in cell proliferation and viability [49,50]. Isolated lung CSCs receiving PS at a concentration of 20 µM alone did not indicate any morphological changes when compared with its respective control. The cells presented with a dense monolayer of cells that are viable with no membrane damage or signs of cytotoxicity; thus, indicating the safety of using AlPcS$_4$Cl as a PS [23,51]. CSCs that received PDT treatment showed

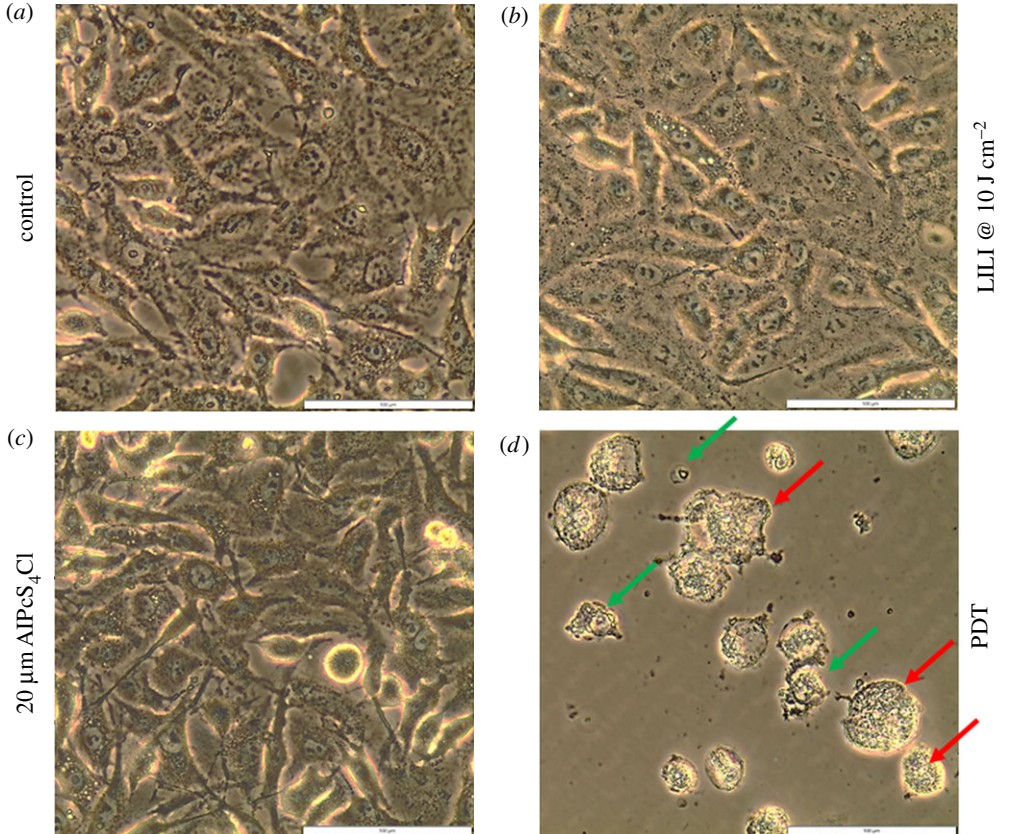

**Figure 5.** Morphology of isolated lung CSCs post-PDT. Cells receiving (*a*) no treatment, (*b*) irradiation of 10 J cm$^{-2}$ and (*c*) PS of 20 µM alone represents healthy viable cells with no cytotoxic morphological changes. (*d*) Lung CSCs that received PDT using 20 µM AlPcS$_4$Cl and 10 J cm$^{-2}$ irradiation show indications of apoptotic (green) and necrotic (red) cell death.

**Table 3.** AlPcS$_4$Cl-PDT dose–response on A549 lung cancer cells. An [IC50]* was achieved using 20 µM AlPcS$_4$Cl and a fluence of 10 J cm$^{-2}$ [23].

| dose AlPcS$_4$Cl (µM): fluence (J cm$^{-2}$) | cytotoxicity (%) | proliferation (%) | viability (%) |
|---|---|---|---|
| 0 : 5 | 0.904532 | 100 | 90.33333 |
| 5 : 5 | 11.98658 | 70.83903 | 89 |
| 10 : 5 | 20.0843 | 33.1369 | 82 |
| 20 : 5 | 24.61926 | 14.15428 | 69 |
| 0 : 10 | 0.922992 | 100 | 90.33333 |
| 5 : 10 | 5.48257 | 84.23562 | 81.66667 |
| 10 : 10 | 23.35169 | 44.45995 | 63.66667 |
| 20 : 10 | *24.94538 | *30.52207 | *42 |
| 0 : 15 | 5.07769 | 100 | 91 |
| 5 : 15 | 12.40134 | 105.5512 | 79.66667 |
| 10 : 15 | 37.60235 | 79.72145 | 66.66667 |
| 20 : 15 | 39.01121 | 60.12904 | 33 |

alterations in morphology, including condensation, fragmentation, apoptotic bodies and vacuolization indicative of apoptosis; cell swelling and lysis were also noted, indicative of necrosis.

### 3.4.2. Cytotoxicity

To explore the PDT activity of AlPcS$_4$Cl on isolated lung CSCs, a series of LDH cytotoxicity assays were performed. Results seen in figure 6 indicated that there was no significant cytotoxicity associated with

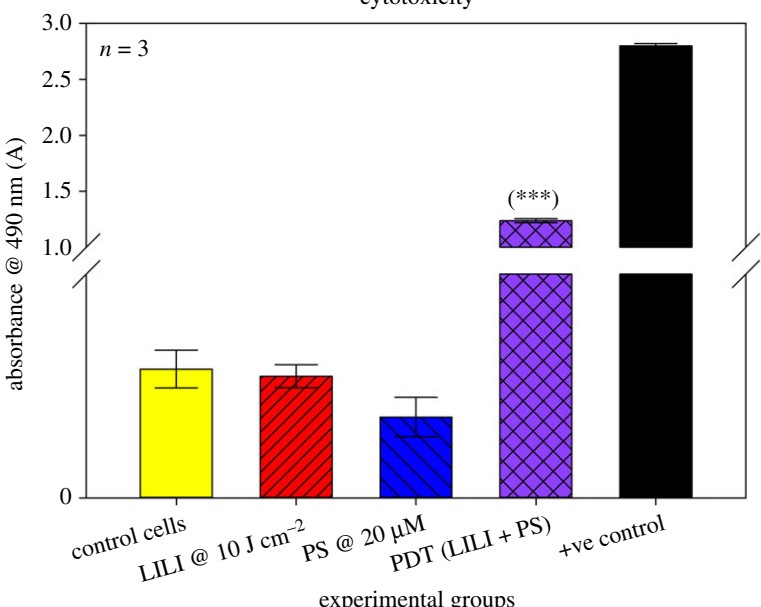

**Figure 6.** LDH cytotoxicity of isolated lung CSCs post-PDT. Cytotoxicity was measured as an absorbance value at 490 nm. All test samples were compared with their respective control cells. No statistical significance was seen when exposing the cells to PS or irradiation alone. A statistical significance of $p < 0.001$ (***) in LDH cytotoxicity was observed when treating the cells with PDT using 20 µM AlPcS$_4$Cl and 10 J cm$^{-2}$ irradiation compared with the control, PS and LILI alone.

exposure to light irradiation alone in the absence of the PS. This result is due to red photo-biomodulation which is used at a low power and low fluence output having a bio stimulatory effect on cells [52,53], thus having no indication of cytotoxicity. The results seen from the group treated with PS alone in the absence of excitation light had shown no significant cytotoxicity. One of the characteristics of an ideal PS is to have negligent dark toxicity [54], which was seen when using AlPcS$_4$Cl [23]. A significant increase in cytotoxicity was observed in CSCs receiving PDT treatment ($p < 0.001$ (***)); this using a pre-established dose [IC50] on lung cancer of 20 µM AlPcS$_4$Cl and 10 J cm$^{-2}$ [23].

### 3.4.3. Proliferation

All metabolically active cells require energy to perform cellular processes such as proliferation. Most cancerous cells produce ATP energy through aerobic glycolysis [55]. To determine whether PDT had an effect on isolated lung CSCs, we measured the amount of ATP present in the test samples in order to determine what amount of ATP is present in metabolically functional lung CSCs and whether PDT reduces cellular processes, which can be seen by a decrease in intracellular ATP measured. Proliferation results (figure 7) indicate that cells treated with irradiation and PS only had no significant impact when compared with their control cells receiving no treatment. Although previous studies established that irradiation at wavelengths between 630 and 800 nm using low energy densities of 10 J cm$^{-2}$ had a proliferative effect on various cell lines [56–58], it was noted that irradiation had no significant effect on lung CSCs. This is in accordance with research stating that CSCs or stem cells have a decreased or quiescent metabolism [59], which ultimately impacts the metabolic rate and the effect irradiation has on lung CSCs, also considering the fluence and wavelength used to irradiate the cells. Due to AlPcS$_4$Cl's negligent dark toxicity, it is noted that the CSCs receiving only PS had a similar metabolic rate as its control cells, due to the PS having no effect in its unactivated state. Lung CSCs that were treated with PDT showed a significant ($p < 0.001$ (***)) decrease in cell metabolism, seen by a decrease in ATP luminescence. Indicating when using an [IC50] previously established on lung cancer cells, 20 µM AlPcS$_4$Cl and 10 J cm$^{-2}$ irradiation, it can significantly decrease the proliferation rate of lung CSCs via metabolic impairment and lead to CSC death *in vitro*.

### 3.4.4. Viability

The Trypan blue exclusion assay identifies viable and dead cells, as it reflects the viability based on cell membrane integrity [60]. Viable cells that have not undergone any stress induction will exclude the dye.

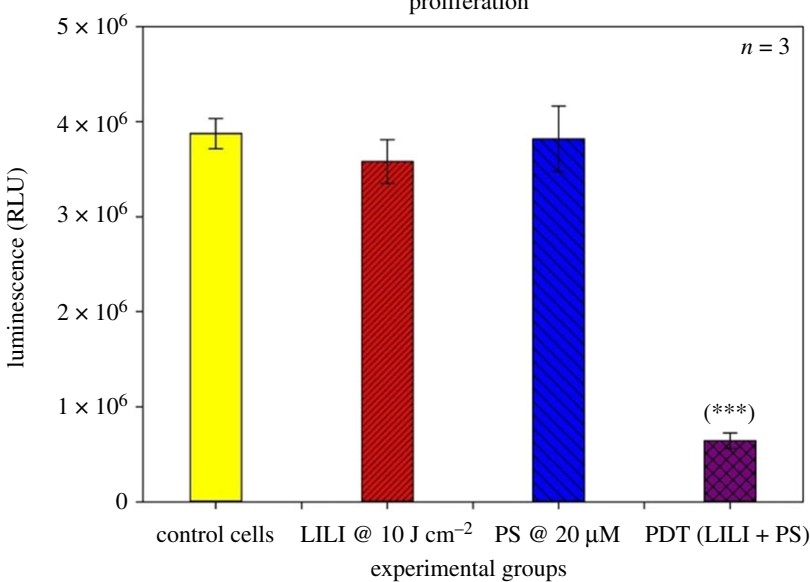

**Figure 7.** ATP proliferation of isolated lung CSCs post-PDT. ATP was measured as a luminescent value in RLUs. All test samples were compared with their respective control cells. No statistical significance was seen when exposing the cells to PS or irradiation alone. A statistically significant decrease in ATP proliferation of $p < 0.001$ (***) was observed when treating the cells with PDT using 20 μM AlPcS$_4$Cl and 10 J cm$^{-2}$ irradiation.

Cells exposed to treatments or environmental factors inducing stress will have diminished cell membrane integrity making the cell permeable to the dye, in the case of PDT, ROS-mediated damage. Results were recorded as a percentage value of the proportion of viable cells after experimental treatment as seen in figure 8. The results show that there is no significant difference in viability in CSCs treated with irradiation and PS alone, as compared with the control sample. Confirming that irradiation at 10 J cm$^{-2}$ does not have a significant stimulatory action on lung CSCs, as well as AlPcS$_4$Cl having no dark toxicity when exposing the cells to the PS without light activation. There was a significant ($p < 0.01$ (**)) decrease seen in viability after PDT treatment of the lung CSCs using the [IC50] obtained from the treatment of the parent lung cancer cell line [23].

### 3.4.5. Mitochondrial membrane potential

Mitochondria exert both vital and lethal functions such as integrate death signals and importantly provide energy to sustain the metabolic needs of cells. Mitochondria membrane potential (Δ$\psi$m), which reflects the mitochondria functional status, is thought to correlate with the cell differentiation status, tumorigenicity and malignancy. The Δ$\psi$m is related to apoptosis due to the fact that dissipation of Δ$\psi$m is a critical event in the apoptosis process [61]. To establish whether AlPcS$_4$Cl was effective at depleting the lung CSCs of their energy source and tumorigenicity, through mitochondrial damage and membrane disruption leading to apoptotic cell death, MitoRed was used to indicate the status of the mitochondria post-irradiation. Results show (figure 9) that CSCs treated with irradiation or PS alone had no mitochondrial damage when compared with the untreated control cells. Lung CSCs receiving PDT showed clear mitochondrial membrane disruption indicated by a loss of clear mitochondrial structure and a decrease in FL intensity.

### 3.4.6. Cell death mechanism

It has been established that PDT directly destroys cancer cells by inducing either apoptotic or necrotic cell death where these mechanisms can occur concurrently [62]. Photodynamic-induced cell death is seen due to the formation of ROS associated with oxidative stress and subsequent cell damage by oxidizing and degrading cell components [63]. To establish whether PDT induced apoptotic or necrotic cell death in isolated lung CSCs, an Annexin V PI cell death assay was used. In apoptotic cells, phosphatidylserine is translocated from the internal to the external part of the plasma membrane, exposing

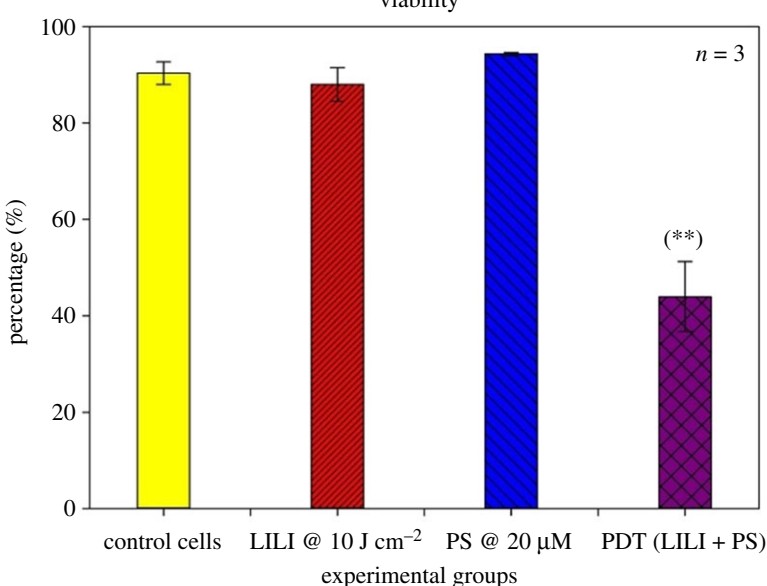

**Figure 8.** Trypan blue viability of isolated lung CSCs post-PDT. Trypan blue was used as a dye exclusion assay, where viable cells excluding the dye were recorded as a percentage value. All test samples were compared with their respective control cells. No statistical significance was seen when exposing the cells to PS or irradiation alone. A statistically significant decrease $p < 0.01$ (**) in viability was seen when treating the cells with PDT using 20 μM AlPcS$_4$Cl and 10 J cm$^{-2}$ irradiation, when compared with the respective control samples.

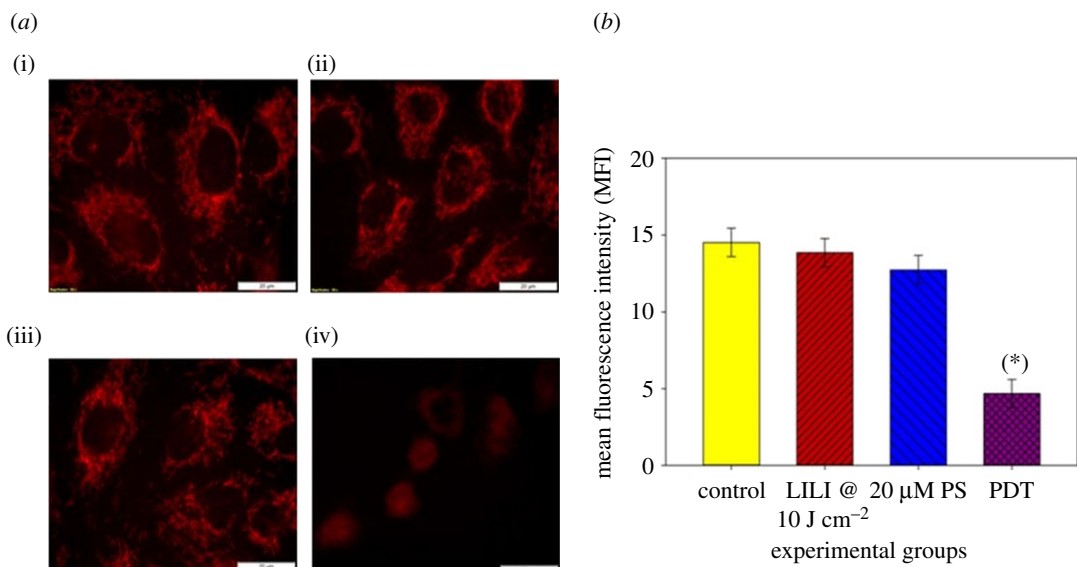

**Figure 9.** Mitochondria membrane potential ($\Delta\psi$m) of isolated lung CSCs post-PDT. (*a*) MitoRed was used to stain mitochondria, where intact mitochondria show a bright red FL and damaged organelles show decreased FL. (i) Lung CSC control, (ii) cells receiving irradiation alone, (iii) cells receiving PS alone without photoactivation and (iv) PDT-treated lung CSCs. Fluorescent intensity was recorded, and all experiments were compared with the control. (*b*) No statistical significance was seen when exposing the cells to PS or irradiation alone. A statistically significant decrease $p < 0.05$ (*) in $\Delta\psi$m was seen when treating the cells with PDT using 20 μM AlPcS$_4$Cl and 10 J cm$^{-2}$ irradiation, when compared with the respective control samples.

phosphatidylserine, making it available for Annexin V, a phospholipid-binding protein, to attach to it [64]. Phosphatidylserines' translocation is an initial occurrence preceding the loss of membrane integrity associated with later cell death pathways. For this reason, PI is used in conjunction with Annexin V, as a nucleic acid intercalator that penetrates porous cell membranes, enabling it to attach to DNA. Making it possible to distinguish between cells that are viable, cells that are in early apoptosis, late apoptosis and necrotic [65]. Flow cytometric results (figure 10) show that PDT significantly increased early apoptosis, late apoptosis and necrosis in lung CSCs.

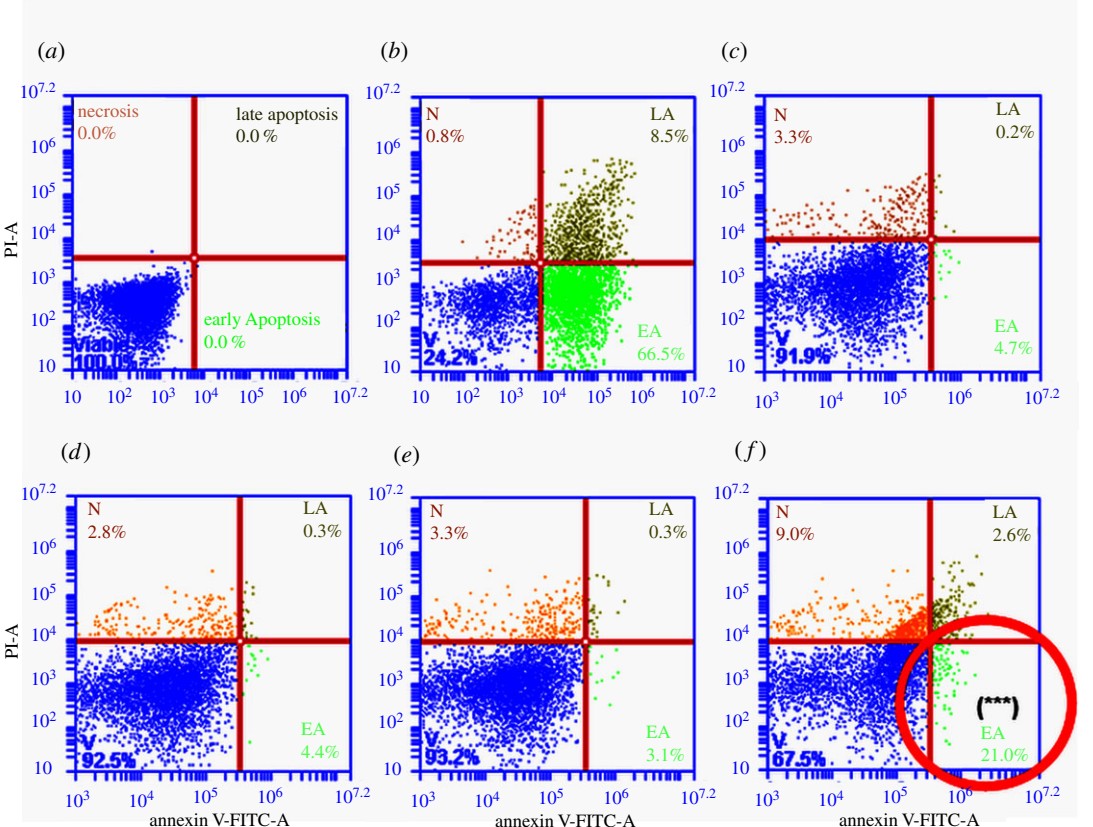

**Figure 10.** Cell death–annexin VPI of isolated lung CSCs post-PDT. Unstained cells were used to establish the cell population (*a*). A positive control sample was used where apoptosis was induced using hydrogen peroxide (*b*). Cells receiving no treatment were used as the control (*c*). Cells receiving PS or irradiation alone had no significant cell death with viabilities of 92.5% and 93.2%, respectively (*d,e*). Cells receiving PDT showed a statistically significant increase $p < 0.001$ (\*\*\*) in early apoptosis (21%), late apoptosis (2.6%) and necrosis (9%) (*f*).

# 4. Discussion

At present, PDT using different types of PSs has been evaluated on various tumour types both *in vitro* and *in vivo* [66]. However, the outcomes of PDT on isolated lung CSCs have yet to be elucidated. In this study, we elaborated on the effects that PDT using AlPcS$_4$Cl has on isolated lung CSCs to establish its treatment potential for lung cancer. Morphologic and biochemical assessments were performed on several groups of isolated lung CSCs displaying stem-like markers including CD133, CD44 and CD56, all of which have been identified to promote cancer recurrence and metastasis. Where CD133 is the most frequently found CSC marker. This cell surface glycoprotein aids in self-renewal, tumour initiation and drug resistance where it has been shown to maintain cancer stem-like and chemo, radio-resistant properties in lung cancer-derived CD133+ cells [67]. CD44 has been reported to be associated with a poor prognosis of lung cancer, where it has been established that lung cancer cell lines that are CD44+ had increased proliferative and colony-forming potential [68]. Neural cell adhesion molecule CD56 has been reported to be highly expressed in cases of lung cancer recurrence, where this marker has been established as a diagnostic tool for lung cancer as well as potential target for drug immunotherapy [69]. Due to the severity of tumorigenic potential seen when lung CSCs express these markers, we used them as a treatment population in our experiments. The lung CSCs were successfully isolated using magnetic cell sorting and characterized by FL and flow cytometric protein detection.

AlPcS$_4$Cl has been used due to its preferred characteristics for an effective PS, as it has been shown to be photo-stable and amphipathic, has no dark cytotoxicity, localizes in intracellular organelles that cause apoptotic cell death due to membrane damage and ROS formation, follows a dose-dependent increase in cell destruction and has an ideal wavelength of activation to reach tissue depth [23]. Mitochondria-targeted agents, such as AlPcS$_4$Cl, are significantly more effective than other apoptosis activating agents for both CSCs and cancer cells, by inducing mitochondrial dysfunction, leading to disruption in cell homeostasis [44].

This is important as lung cancer cells, specifically A549 cell line, are shown to have a higher $\Delta\psi$m than other normal pulmonary cells and to be more prone to proliferation and tumour formation. Even more so, lung CSCs from the A549 lung cancer cell line proved to possess a higher $\Delta\psi$m, where the detection of the CD133 biomarker is detected in these higher $\Delta\psi$m cells. It has been reported that the enhanced tolerance of CSCs to the chemo and radiation correlated well with the changes of $\Delta\psi$m. The $\Delta\psi$m is related to apoptosis, due to the fact that dissipation of $\Delta\psi$m is a critical event in the apoptosis process [70]. Interfering with the intracellular redox balance of CSCs is therefore an important approach to CSC removal [71–73].

Results confirmed that the intracellular confinement of this PS in lung CSCs is optimal for cell death induction, where the PS localizes in the cytosol and around vital intracellular organelles that upon photoactivation produce free radicals which ultimately destroyed these cellular structures inhibiting their functions leading to cell death. PDT treatment results indicated that neither individual components induced any significant changes to the lung CSCs. PDT had significant changes when analysing the morphological and biochemical assessments. Changes in morphology were indicative of apoptotic and necrotic cell death induction caused by the activated PS-forming cytotoxic species that interact with cellular components causing the observed cell damages. Morphological results were confirmed by LDH cytotoxicity, which measured the number of enzymes released into the culture medium upon cell destruction. Results indicated a significant increase in LDH cytotoxicity. Results were further corroborated by ATP proliferation and Trypan blue viability analysis showing a significant decrease in CSC proliferation and viability as compared with the control samples. When comparing results with those established in a previous study using a similar PS and laser parameters on lung cancer cells, it was noted that the amount of cytotoxicity achieved was higher in lung CSCs than lung cancer alone. Similarly, the findings showed a greater decrease in proliferation for CSCs than lung cancer, viability comparisons indicated a similar result in the viability of 43% [23]. Further validating the above-mentioned findings of successful lung CSC destruction was the significant decrease in $\Delta\psi$m, and cell death analysis using Annexin V PI, confirming cell death via apoptosis and necrosis. This would suggest that AlPcS$_4$Cl shows the desired effect when used on lung CSCs as a photodynamic drug. This can be ascribed to the cell death mechanisms induced by PDT, that is the generation of free radicals causing metabolic imbalances in the CSCs leading to cell membrane damage and cell death, where the PS reacts with biomolecules, transferring hydrogen atoms via the radical mechanism generating free radicals and radical ions that react with intracellular oxygen resulting in ROS generation and/or the PS reacting with oxygen in its triplet ground state yielding highly reactive and cytotoxic singlet oxygen. ROS and singlet oxygen have high reactivity and a short half-life. Due to this, PDT directly affects only those biological substrates that are close to the region where these species are generated [74].

Future recommendations for assessing the *in vitro* effects of PDT using AlPcS$_4$Cl on CSCs will include investigating the PDT effects on various cell lines and CSCs from solid tumours, cell migration and invasion assays along with the development of PDT-induced molecular pathways to address any limitations observed in this report. Cancer metastases and the risk of secondary tumours are the most frequent causes of mortality in many cases. One important feature of metastases is the invasive ability of the cells, which is driven primarily by cell motility [75]. Therefore, considering CSC inhibition of migration associated with metastases can be as important as inhibition of CSC proliferation. Different physiological studies including wound healing, transwell cell migration and invasion assay, microfluidic chamber assay and cell exclusion zone assay can be conducted to determine whether PDT prevents CSC migration and invasion [76]. The most important element deciding the outcome of PDT is how a PS interacts with target tissue or tumour cells, and the main feature of this relationship is the subcellular localization of a PS [77]. The exact mechanisms of apoptotic cell death caused by PDT in lung CSCs are still unclear as cell death can be triggered by distinct, yet overlapping, signalling pathways that respond to specific stimuli [78]. The cell death pathways and signalling mechanisms can be explored further through RT-PCR caspase signalling detection, Cytochrome C, ROS detection and DNA damage [79]. Furthermore, photodynamic anti-cancer therapy is aimed at destroying cancerous cells alone, preserving normal cells [80]. Therefore, the effects of the PS need to be explored on normal lung cells before evaluating the effects of using AlPcS$_4$Cl-PDT *in vivo* and clinically to confirm its selectivity and effectivity along with comparing those effects with conventional treatment options and clinically approved PSs.

# 5. Conclusion

CSCs have recently been documented in several cancers and are suggested to prove metastatic ability, relapse and opposition to radiotherapy and chemotherapy. Numerous research demonstrates that tumours include a

specific subgroup of cells that display self-renewal, proliferate seldom, possess multiple pluripotency genes and therefore are accountable for the preservation of tumours and metastases [81]. Cancer treatments that target rapidly proliferating tumour cells do not always impair CSCs, which is why they need to be selectively killed upon treatment to completely eliminate the tumour. Since some, the few that remain unaffected, will be culpable for anti-cancer resistance and flare-up. Several cell surface markers are associated with CSCs, and they can be used to identify only those cells [7]. CSCs can be successfully separated using a variety of techniques, such as flow cytometry and magnetic-associated cell sorting. Intracellular organelles, such as mitochondria, perform essential roles in several biological processes, spanning from cellular respiration to signal transduction and cell death management. Since these mechanisms are important for cancer development, mitochondria have subsequently become an appealing prospect for anti-cancer therapy [45]. Cells with a high $\Delta\psi$m possess a stronger resistance to apoptotic inducers than low $\Delta\psi$m cell populations, where lung CSCs with the CD133 biomarker had proven to evade apoptotic cell death due to increased $\Delta\psi$m. However, this study proves that PDT using AlPcS$_4$Cl has the desired effects of killing lung CSCs, where intracellular localization of the PS was seen to be in the cytosol and around integral organelles, where membrane permeabilization and disruption of cell homeostasis through increased free radical production lead to cellular destruction. This is seen in morphological features of apoptosis and necrosis that included cell shrinkage, chromatin condensation, cell blebbing, cytoplasmic vacuolization and cell swelling and lysis. Furthermore, results on cell toxicity, proliferation and viability corroborated these morphological findings, where AlPcS$_4$Cl-PDT caused a significant increase in cytotoxicity and significant decreases in cell proliferation and viability. AlPcS$_4$Cl-PDT further proved to be effective in causing apoptotic lung CSC death by inducing significant mitochondrial damage shown by a decrease in $\Delta\psi$m, upon photoactivation of the PS causing eradication of the cells through apoptotic cell death. It should be mentioned that PDT can be considered as a palliative treatment along with established lung cancer therapies, which can enhance the prognostic outcome of the treatments by killing of CSCs. CSC eradication should be achieved when using higher doses of PDT, following dose-dependent cell death seen on the parent lung cancer cell line [23]. Studies to further improve the effects of PDT and to make it specific are currently under study where clinical PS toxicity arising from non-specific uptake of PS drugs, along with a general question regarding the limited light penetration used for photoactivation in PDT, is under investigation. PDT can be directly aimed at cancerous cells by using nanoparticle and antibody conjugation of the PS, making it more specific and improving its absorption into cancer cells [28]. PDT may be used clinically with interstitial, endoscopic, intraoperative or laparoscopic light distribution devices with the use of fibres that can be placed at designated positions inside the tumour site due to developments in fibre optics and microendoscopic technology [82].

Data accessibility. The datasets supporting this article have been uploaded to an institutional data repository: https://figshare.com/s/6baad4282697da0080da where the data are available from the following doi:10.25415/ujhb.13643009.

Authors' contributions. Significant contributions were made by both authors, A.C. and H.A. Conceptualization was done by A.C. and H.A.; funding acquisition was done by H.A.; investigation has done by A.C.; methodology was done by A.C.; project administration was done by A.C. and H.A.; resources were arranged by H.A.; supervision was done by H.A.; validation was done by A.C. and H.A.; visualization was done by A.C.; writing the original draft was done by A.C.; writing the review and editing was done by A.C. and H.A.

Competing interests. The authors declare no conflicts of interest.

Funding. This research was supported by the National Research Foundation (NRF) S&F—Scarce Skills Postdoctoral Fellowship (grant no. 120752) received by A.C. and the South African Research Chairs Initiative of the Department of Science and Technology and National Research Foundation of South Africa (SARChI/NRF-DST) (grant no. 98337) received by H.A.

Acknowledgements. The authors sincerely thank the University of Johannesburg and the National Laser Centre for their facilities and laser equipment.

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
