## [Peer Review File · Royal Society Open Science]

Review History

RSOS-210148.R0 (Original submission)

Review form: Reviewer 1

Is the manuscript scientifically sound in its present form?

No

Are the interpretations and conclusions justified by the results?

No

Is the language acceptable?

Yes

Do you have any ethical concerns with this paper?

No

Have you any concerns about statistical analyses in this paper?

Yes

Recommendation?

Major revision is needed (please make suggestions in comments)

Comments to the Author(s)

This manuscript by Crous et al. investigates a photosensitizer (AIPcS4CL) to kill lung cancer stem cells. Following purification/characterisation of lung cancer stem cells the authors proceed to investigate the potency and potential mechanism of action of the photosensitiser, showing that it effectively kills cancer cells. Overall the work is timely, interesting and well performed. However, there are few expts. that in my opinion have been over interpreted, require additional work or toning down of conclusions, these are detailed further.

- Figure 4 - due to the poor quality of the images (and inherent low resolution with conventional IF) it really is impossible to state that the PS is in mitochondria and/or lysosomes, I think the most that can be stated with any certainty is that it (PS) is in the cell, though ideally a non PS incubated cell should be included as negative control for the red channel. Conclusions from this expt. should be toned down.

- Figure 6. statistics - the LDH assay states an N=3, its difficult (near impossible) to consider these data as biological replicates given the extremely small error bars, are these technical replicates ? - if so the text requires modification and some indication of the reproducibility of the expts should be provided.

- Figure 5/Figure 10, cells with PS plus light don't look apoptotic by morphology - does caspase inhibition prevent or at least slow the extent of cell death ? I think this is necessary to conclude the type of cell death is apoptotic.

- Figure 9. Loss of mitochondrial membrane potential invariably occurs during cell death, authors nicely demonstrate loss of mitochondrial membrane potential but its frankly impossible to state that this is causative of cell death - the text should be modified accordingly

- title should be changed to encompass the authors' findings not the purpose/question of the study

Review form: Reviewer 2

Is the manuscript scientifically sound in its present form?

Yes

Are the interpretations and conclusions justified by the results?

Yes

Is the language acceptable?

Yes

Do you have any ethical concerns with this paper?

No

Have you any concerns about statistical analyses in this paper?

No

Recommendation?

Accept as is

Comments to the Author(s)

This MS describes the photosensitizing properties of AIPcS₄Cl for targeting CSCs in lung cancers as they are considered responsible for the cell proliferation, cancer recurrence, metastasis, and resistance of cancer to drugs.

Various cell biology methodologies have been used to demonstrate, with regularly positive results, the efficacy of a combination of the new PS at 20 uM plus red light (10 J/cm²).

Although the idea of this study and the experiments are not so innovative, I think this is a good MS, well written, with good figures and explanations, and more importantly, the results could be useful for the palliative treatment of lung cancer with AIPcS₄Cl-PDT.

Decision letter (RSOS-210148.R0)

Dear Dr Crous,

The Editors assigned to your paper RSOS-210148 "Photodynamic Therapy for Lung Cancer Stem Cell Eradication - Is AIPcS₄Cl an Effective Photosensitizer?" have now received comments from reviewers and would like you to revise the paper in accordance with the reviewer comments and any comments from the Editors. Please note this decision does not guarantee eventual acceptance.

Please submit your revised manuscript and required files (see below) no later than 21 days from today's (ie 28-Jun-2021) date. Note: the ScholarOne system will 'lock' if submission of the revision is attempted 21 or more days after the deadline. If you do not think you will be able to meet this deadline please contact the editorial office immediately.

on behalf of Dr Simon Cook (Associate Editor) and Catrin Pritchard (Subject Editor)
openscience@royalsociety.org

Associate Editor Comments to Author (Dr Simon Cook):

Your manuscript has now been reviewed. The manuscript is considered to be of interest to the readership of Open Science. However, it is felt that some of your conclusions are premature or over-interpreted.

Please in particular consider the comments of Reviewer 1 below in preparing a revised version of your manuscript

- Figure 4 - due to the poor quality of the images (and inherent low resolution with conventional IF) it really is impossible to state that the PS is in mitochondria and/or lysosomes, I think the most that can be stated with any certainty is that it (PS) is in the cell, though ideally a non PS incubated cell should be included as negative control for the red channel. Conclusions from this expt. should be toned down.

- Figure 6. statistics - the LDH assay states an N=3, its difficult (near impossible) to consider these data as biological replicates given the extremely small error bars, are these technical replicates ? - if so the text requires modification and some indication of the reproducibility of the expts should be provided.

- Figure 5/Figure 10, cells with PS plus light don't look apoptotic by morphology - does caspase inhibition prevent or at least slow the extent of cell death ? I think this is necessary to conclude the type of cell death is apoptotic.

- Figure 9. Loss of mitochondrial membrane potential invariably occurs during cell death, authors nicely demonstrate loss of mitochondrial membrane potential but its frankly impossible to state that this is causative of cell death - the text should be modified accordingly

- title should be changed to encompass the authors' findings not the purpose/question of the study

I look forward to receiving your revised manuscript

Reviewer comments to Author:

Reviewer: 1

Comments to the Author(s)

This manuscript by Crous et al. investigates a photosensitizer (AIPcS4CL) to kill lung cancer stem cells. Following purification/characterisation of lung cancer stem cells the authors proceed to investigate the potency and potential mechanism of action of the photosensitiser, showing that it effectively kills cancer cells. Overall the work is timely, interesting and well performed. However, there are few expts. that in my opinion have been over interpreted, require additional work or toning down of conclusions, these are detailed further.

- Figure 4 - due to the poor quality of the images (and inherent low resolution with conventional IF) it really is impossible to state that the PS is in mitochondria and/or lysosomes, I think the most that can be stated with any certainty is that it (PS) is in the cell, though ideally a non PS incubated cell should be included as negative control for the red channel. Conclusions from this expt. should be toned down.

- Figure 6. statistics - the LDH assay states an N=3, its difficult (near impossible) to consider these data as biological replicates given the extremely small error bars, are these technical replicates ? - if so the text requires modification and some indication of the reproducibility of the expts should be provided.

- Figure 5/Figure 10, cells with PS plus light don't look apoptotic by morphology - does caspase inhibition prevent or at least slow the extent of cell death ? I think this is necessary to conclude the type of cell death is apoptotic.

- Figure 9. Loss of mitochondrial membrane potential invariably occurs during cell death, authors nicely demonstrate loss of mitochondrial membrane potential but its frankly impossible to state that this is causative of cell death - the text should be modified accordingly

- title should be changed to encompass the authors' findings not the purpose/question of the study

Reviewer: 2

Comments to the Author(s)

This MS describes the photosensitizing properties of AlPcS4Cl for targeting CSCs in lung cancers as they are considered responsible for the cell proliferation, cancer recurrence, metastasis, and resistance of cancer to drugs.

Various cell biology methodologies have been used to demonstrate, with regularly positive results, the efficacy of a combination of the new PS at 20 uM plus red light (10 J/cm²).

Although the idea of this study and the experiments are not so innovative, I think this is a good MS, well written, with good figures and explanations, and more importantly, the results could be useful for the palliative treatment of lung cancer with AlPcS4Cl-PDT.

===PREPARING YOUR MANUSCRIPT===

===PREPARING YOUR REVISION IN SCHOLARONE===

<https://royalsociety.org/journals/authors/author-guidelines/#data>. You should ensure that

you cite the dataset in your reference list. If you have deposited data etc in the Dryad repository, please include both the 'For publication' link and 'For review' link at this stage.

Author's Response to Decision Letter for (RSOS-210148.R0)

See Appendix A.

Decision letter (RSOS-210148.R1)

Dear Dr Crous,

It is a pleasure to accept your manuscript entitled "AlPcS₄Cl is an Effective Photosensitizer for the Eradication of Lung Cancer Stem Cells" in its current form for publication in Royal Society Open Science. The comments of the reviewer(s) who reviewed your manuscript are included at the foot of this letter.

===COVID-SPECIFIC TEXT -- WILL ONLY BE ADDED TO COVID-PAPERS BY THE EDITORIAL OFFICE===

COVID-19 rapid publication process:

We are taking steps to expedite the publication of research relevant to the pandemic. If you wish, you can opt to have your paper published as soon as it is ready, rather than waiting for it to be published the scheduled Wednesday.

This means your paper will not be included in the weekly media round-up which the Society sends to journalists ahead of publication. However, it will still appear in the COVID-19 Publishing Collection which journalists will be directed to each week (<https://royalsocietypublishing.org/topic/special-collections/novel-coronavirus-outbreak>).

If you wish to have your paper considered for immediate publication, or to discuss further, please notify openscience_proofs@royalsociety.org and press@royalsociety.org when you respond to this email.

===END OF COVID-SPECIFIC TEXT -- WILL BE REMOVED AS NECESSARY BY THE EDITORIAL OFFICE===

on behalf of Dr Payam Gammage (Associate Editor) and Catrin Pritchard (Subject Editor)
openscience@royalsociety.org

Associate Editor Comments to Author (Dr Payam Gammage):
Associate Editor
Comments to the Author:
(There are no comments.)

Reviewer comments to Author:

13 July 2021

To: Dr Simon Cook (Associate Editor) and Catrin Pritchard (Subject Editor)

Title: " AIPcS₄Cl is an Effective Photosensitizer for the Eradication of Lung Cancer Stem Cells"

Authors: Dr Anine Crous and Prof Heidi Abrahamse

CORRECTIONS TO MANUSCRIPT: RSOS-210148

Dear Dr Simon Cook

Thank you for the opportunity to strengthen the manuscript Titled: "**AIPcS₄Cl is an Effective Photosensitizer for the Eradication of Lung Cancer Stem Cells**" as suggested by the reviewers. The reviewers provided informative feedback and recommendations. All statements were addressed, and revisions were made for publishing in the *Royal Society Open Science*.

Please find below the reports of the reviewers, together with the comments and corrections made, as shown in red.

I trust these corrections will meet with your approval.

Thank you for your support.

Sincerely,

Dr Anine Crous (PhD)
Postdoctoral Fellow
Laser Research Centre
Faculty of Health Sciences
University of Johannesburg
South Africa
Email: acrous@uj.ac.za

RESPONSE TO REVIEWERS: RSOS-210148

Associate Editor:

"Your manuscript has now been reviewed. The manuscript is considered to be of interest to the readership of Open Science. However, it is felt that some of your conclusions are premature or over-interpreted. Please in particular consider the comments of Reviewer 1 below in preparing a revised version of your manuscript. I look forward to receiving your revised manuscript."

Comment and correction:

The authors would like to thank the editor for their feedback and the opportunity to revise and improve the manuscript as suggested by reviewer 1.

Reviewer #1:

This manuscript by Crous et al. investigates a photosensitizer (AIPcS4CL) to kill lung cancer stem cells. Following purification/characterisation of lung cancer stem cells the authors proceed to investigate the potency and potential mechanism of action of the photosensitiser, showing that it effectively kills cancer cells. Overall the work is timely, interesting and well performed. However, there are few expts. that in my opinion have been over interpreted, require additional work or toning down of conclusions, these are detailed further.

Comment and correction:

The authors would like to thank the reviewer for their positive feedback, the manuscript was revised where overinterpretation of experiments and conclusions were toned down as seen below.

- Figure 4 - due to the poor quality of the images (and inherent low resolution with conventional IF) it really is impossible to state that the PS is in mitochondria and/or lysosomes, I think the most that can be stated with any certainty is that it (PS) is in the cell, though ideally a non PS incubated cell should be included as negative control for the red channel. Conclusions from this expt. should be toned down.

Comment and correction:

Figure 4. Image processed using Image J, where backgrounds were subtracted to enhance the FL images.

Conclusions from intracellular localisation of the PS toned down:

Page 9; line 283-285: Results: "Results show AIPcS4Cl is embedded in the cytosol and perinuclear region, that upon photoactivation can cause peripheral photodamage to the cell membrane and plasma membranes of various intracellular organelles including mitochondria and lysosomes."

Page 17; line 456: Discussion: "...where the PS localizes in the cytosol and around vital intracellular organelles..."

- Figure 6. statistics - the LDH assay states an N=3, its difficult (near impossible) to consider these data as biological replicates given the extremely small error bars, are these technical replicates ? - if so the text requires modification and some indication of the reproducibility of the expts should be provided.

Comment and correction:

Figure 6. An axial break (Y-axis) was included in the graph for ease of viewing experimental errors. All experimental groups were performed 3 times (n=3) (biological replicates). Furthermore, all spectroscopy assays were performed in duplicate (technical replicates).

- Figure 5/Figure 10, cells with PS plus light don't look apoptotic by morphology - does caspase inhibition prevent or at least slow the extent of cell death ? I think this is necessary to conclude the type of cell death is apoptotic.

Comment and correction:

Morphological findings seen and described motivate for the apoptotic and necrotic cell death conclusions and were edited as below. Furthermore, the accumulated experimental results specifically, mitochondrial membrane potential, and Annexin VPI cell death mechanisms indicate apoptotic and necrotic cell death.

Page 10; line 312 – 313: Results: "...including, condensation, fragmentation, apoptotic bodies and vacuolization indicative of apoptosis, cell swelling and lysis was also noted indicative of necrosis..."

Page 1; line 317-318: Figure 5 Caption: "...d) Lung CSCs that received PDT using 20 μ M AlpcS4Cl and 10J/cm² irradiation, show indications of apoptotic (green) and necrotic (red) cell death."

- Figure 9. Loss of mitochondrial membrane potential invariably occurs during cell death, authors nicely demonstrate loss of mitochondrial membrane potential but its frankly impossible to state that this is causative of cell death - the text should be modified accordingly

Comment and correction:

The authors are not aware of the above statement: "los of MMP is caused by cell death". The authors checked that the correct statement referring to 'loss of MMP **leads** to cell death' was used throughout the text.

- title should be changed to encompass the authors' findings not the purpose/question of the study

Comment and correction:

The title was changed to encompass the research findings: "AIPcS4Cl is an Effective Photosensitizer for the Eradication of Lung Cancer Stem Cells."

Reviewer #2:

This MS describes the photosensitizing properties of AIPcS4Cl for targeting CSCs in lung cancers as they are considered responsible for the cell proliferation, cancer recurrence, metastasis, and resistance of cancer to drugs. Various cell biology methodologies have been used to demonstrate, with regularly positive results, the efficacy of a combination of the new PS at 20 μ M plus red light (10 J/cm²). Although the idea of this study and the experiments are not so innovative, I think this is a good MS, well written, with good figures and explanations, and more importantly, the results could be useful for the palliative treatment of lung cancer with AIPcS4Cl-PDT.

Comment:

The authors would like to thank the reviewer for their time in reviewing the manuscript and the overall positive comments regarding the research.